# Effectiveness and Safety of Direct Oral Anticoagulants versus Vitamin K Antagonists for People Aged 75 Years and over with Atrial Fibrillation: A Systematic Review and Meta-Analyses of Observational Studies

**DOI:** 10.3390/jcm8040554

**Published:** 2019-04-24

**Authors:** Anneka Mitchell, Margaret C. Watson, Tomas Welsh, Anita McGrogan

**Affiliations:** 1Department of Pharmacy and Pharmacology, University of Bath, Bath BA2 7AY, UK; magswatsonbusiness@gmail.com (M.C.W.); a.mcgrogan@bath.ac.uk (A.M.G.); 2Pharmacy Research Centre, University Hospital Southampton, Southampton SO16 6YD, UK; 3Research Institute for the Care of Older People (RICE), Bath BA1 3NG, UK; tomas.welsh@nhs.net; 4Institute of Clinical Neurosciences, University of Bristol, Bristol BS8 1TH, UK

**Keywords:** anticoagulants, atrial fibrillation, stroke, hemorrhage, aged

## Abstract

Older people, are underrepresented in randomised controlled trials of direct oral anticoagulants (DOACs) for stroke prevention in atrial fibrillation (AF). The aim of this study was to combine data from observational studies to provide evidence for the treatment of people aged ≥75 years. Medline, Embase, Scopus and Web of Science were searched. The primary effectiveness outcome was ischaemic stroke. Safety outcomes were major bleeding, intracranial haemorrhage, gastrointestinal bleeding, myocardial infarction, and mortality. Twenty-two studies were eligible for inclusion. Two studies related specifically to people ≥75 years but were excluded from meta-analysis due to low quality; all data in the meta-analyses were from subgroups. The pooled risk estimate of ischaemic stroke was slightly lower for DOACs. There was no significant difference in major bleeding, mortality, or myocardial infarction. Risk of intracranial haemorrhage was 44% lower with DOACs, but risk of GI bleeding was 46% higher. Our results suggest that DOACs may be preferable for the majority of older patients with AF, provided they are not at significant risk of a GI bleed. However, these results are based entirely on data from subgroup analyses so should be interpreted cautiously. There is a need for adequately powered research in this patient group.

## 1. Introduction

The prevalence of atrial fibrillation (AF) increases with age, with 14% of over 80s having this condition [1]. AF predisposes sufferers to the risk of embolic stroke, and it is estimated that 25% of all strokes in those over 80 years occur in patients with AF [1]. Strokes due to AF tend to be larger, more severe and result in higher mortality rates than their non-AF counterparts [2].

Meta-analyses have consistently shown that among patients with AF, with a moderate to high risk of thromboembolic events, anticoagulation with warfarin, a vitamin K antagonist (VKA), significantly reduces the incidence of stroke with an acceptable bleeding risk compared with placebo [3]. Trials in older people comparing warfarin with aspirin also found this to be the case [4], yet individuals over 80 years old are less likely to be anticoagulated with warfarin than younger people even where their stroke and bleeding risks are the same [5].

The direct oral anticoagulants (DOACs), dabigatran, rivaroxaban, apixaban and edoxaban, were introduced from 2010 onwards as alternatives to VKAs. The initiation rate of DOACs increased 17-fold between 2012 and 2015 in UK general practice and use of VKAs has subsequently declined [6]. DOACs have been recommended internationally as an option for stroke prevention in AF for people with additional risk factors for stroke [7,8]. Additional risk factors are defined by the CHA_2_DS_2_-VASc score as chronic heart failure, hypertension, age ≥75 years, diabetes mellitus, prior stroke/transient ischaemic attack, vascular disease, age 65–74 years, female sex. This risk stratification tool defines being aged ≥75 years as a major risk factor for stroke and allocates a score of 2 to this risk factor, meaning that all patients in this age group should be considered for anticoagulation.

To date, no randomised controlled trials (RCTs) have included older people (defined as ≥75 years) as the primary population of interest and inclusion and exclusion criteria limit the generalisability of the results to older people. Numerous co-morbidities and medications would cause patients to be ineligible to join the trial, and a number of these such as cancer, cognitive impairment and severe renal impairment are more common in older people. 

Whilst numerous systematic reviews and meta-analyses of DOACs have been published in recent years, none focus solely on observational studies of older people aged ≥75 years where there is a need to better understand effectiveness and safety. The aim of this review and meta-analyses was to combine data from observational studies comparing DOACs with VKAs for people ≥75 years with AF to gain generalisable estimates of the effectiveness and safety of specific outcomes in this population. The outcomes being considered are ischaemic stroke, as a measure of effectiveness; major bleeding, intracranial haemorrhage, gastrointestinal bleeding, mortality and myocardial infarction as measures of safety. The main aim of anticoagulation is to reduce the risk of ischaemic stroke, consequently this was chosen as the primary effectiveness outcome. Major bleeding and the risk of intracranial haemorrhage is a significant concern to those prescribing anticoagulants particularly to older people who may be at higher risk of falls. The risk of gastrointestinal bleeding was shown to increase with DOACs in the RCTs by up to 79% and myocardial infarction increased by 38% with dabigatran in the RE-LY trial.

## 2. Experimental Section

### 2.1. Search Strategy and Selection Criteria

For this systematic review and meta-analyses, we searched without language restrictions for observational studies comparing the use of DOACs with VKAs for older people (defined as aged ≥75 years) with atrial fibrillation. Studies were excluded if results did not specify outcome data for participants aged 75 years and over; if anticoagulation was prescribed for multiple indications and results were not stratified for participants with AF; if a DOAC was being compared with a non-vitamin K antagonist anticoagulant or no anticoagulation. We also excluded studies that used randomised, controlled or interventional study designs.

We searched Medline, Embase, Scopus, and Web of Science from 1st January 2009, when the DOACs were first licensed, to 3rd January 2018. Full search terms and search strategy for both Medline and Embase are presented in Appendix B. Database searches were supplemented by contacting pharmaceutical companies to request unpublished data. Reference lists of relevant studies, reviews, and letters were screened for additional articles. Foreign language articles were translated. 

A.M. conducted the searches and screened all titles and abstracts for inclusion. A.M.G. independently duplicated screening of 10% of titles and abstracts. Full-text review was conducted independently and in duplicate by two reviewers (A.M. and either A.M.G., M.C.W. or M.L.). Conflicts were resolved through discussion.

Studies were included in the systematic review and meta-analyses if they compared at least one DOAC with a VKA and, presented data of one or more outcomes of interest for participants aged ≥75 years with atrial fibrillation. The study protocol was registered with PROSPERO (CRD42018081696) prior to starting the systematic review and is available online [9].

### 2.2. Data Analysis

Data extraction was completed independently and in duplicate by A.M. and A.M.G. using a pre-piloted form in Microsoft Access. Primary effectiveness outcomes of interest were ischaemic stroke and/or systemic embolism. Composite outcome measures of effectiveness (e.g., all strokes, or stroke and transient ischaemic attack or other thromboembolic events) were analysed as secondary outcome measures. Primary safety outcomes were major bleeding, gastrointestinal bleeding and intracranial haemorrhage. Major bleeding was primarily defined as per the International Society on Thrombosis and Haemostasis (ISTHM): “A reduction in haemoglobin of at least 20 g/L, and transfusion of at least 2 units of blood, or symptomatic bleeding in a critical area or organ”. However, similar definitions and hospitalisation for bleeding were also included as major bleeding. Secondary outcomes were: all-cause mortality, non-major or other bleeding, and myocardial infarction and composite safety outcomes (e.g., all bleeding, or any combination of safety outcomes of interest).

The quality of studies was assessed using the Newcastle–Ottawa Scale by duplicate, independent assessment (A.M. and A.M.G.). Discrepancies were resolved through discussion and the studies were rescored using a modified version of the Newcastle–Ottawa scale which better reflected the reviewers’ bias assessment (Appendix C).

Data were extracted for the following variables: total number of participants, total number of participants aged ≥75 years, sex, data source, healthcare setting, co-morbidities and medication at baseline, definition of exposure to DOAC or VKA, dose of DOAC, inclusion and exclusion criteria, hazard ratios or incident rates comparing DOAC to VKA in participants aged over 75 years only (or event numbers if summary estimate not available) for all outcomes of interest.

All studies scoring ≥6 on the modified Newcastle–Ottawa scale were included in the meta-analyses. Adjusted hazard ratios and their 95% confidence intervals were extracted and pooled for meta-analyses, results were stratified by individual DOAC then grouped by DOAC dose and age band. For the purposes of this study DOAC dose was stratified as “low” (dabigatran 75 mg or 110 mg, rivaroxaban 15 mg, apixaban 2.5 mg) or “standard” (dabigatran 150 mg, rivaroxaban 20 mg, apixaban 5 mg). Where hazard ratios were not reported, incident rates were used to calculate incident rate ratios and associated 95% confidence intervals. Heterogeneity was assessed using the Cochrane Q statistic, and Higgins and Thompsons’ I^2^. Heterogeneity was defined as low if I^2^ = 25%, moderate if I^2^ = 50%, or high if I^2^ = 75% [10]. High levels of heterogeneity were anticipated due to the varying designs of observational studies so both a fixed model (using the inverse-variance method) and a random-effects model (using the DerSimonian and Laird method) were used to combine the data. Statistical analysis was undertaken using Stata 14 [11] and the Stata package admetan [12].

## 3. Results

### 3.1. Study Identification

We identified 12,330 records through database searching. Title and abstract screening excluded 12,019 records, a further 289 were excluded after full text screening. Finally, 22 studies were eligible for inclusion in the systematic review [13,14,15,16,17,18,19,20,21,22,23,24,25,26,27,28,29,30,31,32,33,34]. Only 20 studies were included in the meta-analyses as two scored <6 on the Newcastle–Ottawa scale. Hand-searching of reference lists did not identify any additional relevant studies. No data were received from pharmaceutical companies. Full details of the selection process are shown in Figure 1. Of the included studies, only two were specifically designed to investigate outcomes in people aged ≥75 years [22,25]. The remaining studies presented some data for older people, but this was limited to subgroup or sensitivity analyses. 

### 3.2. Characteristics of Included Studies

The 20 studies included in the meta-analysis included over 428,031 patients aged ≥75 years. However, not all studies reported the total number of patients in this age group. Appendix A provides a breakdown of the total participants in each study. Eleven studies were conducted in the USA and Canada [13,16,17,18,24,26,28,29,30,32,34], followed by Asia [14,15,22,25,33] (*n* = 5), Europe [19,20,21,27,31] (*n* = 5), and New Zealand [23] (*n* = 1). Most studies compared dabigatran with a vitamin K antagonist [13,14,15,16,17,18,19,20,21,22,23,24,25,26,27,28,29,30,31,33,34] (*n* = 21); rivaroxaban [14,16,17,18,19,20,21,25,27,28,31] and apixaban [14,17,19,20,21,31,32] were licensed later so have not been studied as extensively (*n* = 11 and *n* = 7, respectively). The VKA comparator was predominantly warfarin; one study allowed any VKA, but the majority of participants were prescribed fluindione [27]. Appendix A shows a summary of the main characteristics of the included studies.

### 3.3. Risk of Bias

The Newcastle–Ottawa scale has been widely used in the literature, but we found that interrater agreement on some domains was low due to the subjective nature of the questions and vague decision rules accompanying the tool, a limitation which has been noted elsewhere [35]. We therefore modified the tool to make it more specific and transparent (Appendix C). Overall, the quality of included studies was high, with over half scoring 8 or more out of 12 [15,16,20,21,23,26,27,29,30,31,32,33]. Two studies scored less than 6 and were excluded from the meta-analyses. Table 1 shows the score for each study. Exclusion of these studies is unlikely to have affected the result of the meta-analyses as they were very small studies [22,25].

### 3.4. Outcomes

Overall, the pooled risk estimate for both ischaemic stroke and composite effectiveness outcomes were of borderline significance (ischaemic stroke: seven studies, 0.86 (95% CI 0.75–0.99), Figure 2; composite effectiveness outcomes: nine studies, 0.93 (95% CI 0.85–1.01), Figure 3) between DOACs and VKA. There were no significant differences between DOACs and VKA in the composite or majority of individual safety outcomes (composite safety: six studies, 0.95 (95% CI 0.87–1.04), Figure 4; major bleed: 12 studies, 0.96 (95% CI 0.84–1.09), Figure 5; mortality: six studies, 0.92 (95% CI 0.77–1.10), Figure 6; myocardial infarction: five studies, 0.97 (95% CI 0.81–1.18), Figure 7). However, there was a significantly lower risk of intracranial haemorrhage (10 studies, 0.56 (95% CI 0.48–0.67), Figure 8) and a significantly higher risk of gastrointestinal bleeding (nine studies, 1.46 (95% CI 1.31–1.63), Figure 9) with DOACs than with VKAs.

Differences were shown when the DOACs were analysed individually. Dabigatran had a lower risk of mortality than VKA (three studies, 0.78 (95% CI 0.61–0.99)). Rivaroxaban performed less favourably than DOACs overall in safety outcomes. There was no significant difference in risk of intracranial haemorrhage (two studies, 0.79 (95% CI 0.41–1.54)), and a higher risk of major bleeding (three studies, 1.17 (95% CI 1.04–1.33)) when compared with VKA. Apixaban had a significantly lower risk of major bleeding (three studies, 0.68 (95% CI 0.55–0.84)), but there was no significant difference in the risk of intracranial haemorrhage (one study, 0.82 (95% CI 0.46–1.46)).

Nielsen and colleagues [19] found an excess risk of death in both the apixaban and rivaroxaban groups (1.54 (95% CI 1.4–1.7), 1.67 (95% CI 1.49–1.87) respectively). This was the only study that looked at mortality with these agents and they only considered low doses of these drugs.

## 4. Discussion

This systematic review with meta-analyses is the first to investigate the effectiveness and safety of DOACs compared with VKAs in people aged 75 years and over. We found that there were no significant differences in effectiveness outcomes between DOACs and VKAs, but their safety outcomes varied. The risk of ischaemic stroke was similar and there was no difference when composite outcomes were analysed. There was no significant difference between DOACs and VKAs for major bleeding. However, for site-specific bleeding, there was a 46% increased risk of gastrointestinal bleeding and a 44% decreased risk of intracranial haemorrhage in the DOAC group compared with the VKA group. In contrast to the combined DOAC results, dabigatran alone was associated with a lower risk of major bleeding. Rivaroxaban had a higher risk of major bleeding. There was a lower risk of major bleeding with apixaban than VKA; no studies investigated gastrointestinal bleeding with apixaban.

The main strengths of this study are the focus on older people who have been underrepresented in the RCTs, the use of real world data which are more representative of older people being treated with anticoagulants in clinical practice, and the broad search strategy (which allowed identification of studies where older people were the primary study population or where they were analysed as a subgroup). The quality of each study included in the review was thoroughly assessed, with emphasis on the methods used, exposure definition, outcomes, bias and the potential for misclassification. We translated foreign language studies to minimise the risk of bias. The primary weakness was the use of aggregated data as opposed to using patient level data. We have not been able to validate the results reported as the authors did not include sufficient information. These meta-analyses highlighted the scarcity of evidence comparing the safety and effectiveness of DOACs with VKAs in people aged ≥75 years with AF. Only two studies looked specifically at outcomes in older people and these studies were small, conducted in Asia and low quality which led to exclusion from the meta-analyses [22,25]. All other data came from subgroup or sensitivity analyses within larger studies, which meant that detailed information was often lacking for this specific group and we had to rely on limited results presented as tables or graphs.

The included studies had a number of strengths. Most included a treatment population and comparator group representative of people with AF. Most studies considered a wide range of potential confounders and accounted for these in the analysis using a variety of methods (propensity scores used for matching [13,15,23,26,27,28,32,34], or weighting of survival models [17,19,20,29,30,33], or as a covariate in the survival model [24,31]; high dimensional propensity scores used for matching [16], or adjustment of survival model [18]; adjustment within survival models for age, sex, risk scores and baseline co-morbidities [20] and baseline medications [21]).

The main weaknesses of the included studies firstly related to how exposure was measured. The majority of studies used prescription records from administrative or claims data. All studies described how the initial exposure was identified, but half did not describe how ongoing exposure was measured [16,18,19,24,29,31,33] or stated that an intention-to-treat approach was used [14,24,27]. Two studies were conducted using registry data from a single hospital and did not describe how exposure was measured [22,25]. Observational studies using routinely collected data have been criticised for not considering variation in compliance. It could therefore be argued that not describing ongoing exposure is worse as there is a further risk of misclassification. This is particularly the case where people may have switched anticoagulant or stopped treatment altogether. Secondly, whilst most studies addressed the influence of confounders, three studies did very little to control for confounding [14,22,25]. Where matching was used to control for confounding, there was sometimes a loss of significant numbers of patients from the VKA group who were not eligible for matching [13,15] which may have limited overall generalisability. Follow-up was relatively short for most studies. Ten studies had less than one year’s follow-up for the DOAC group [13,14,21,25,26,27,30,31,32,33], and four studies did not state the average follow-up time [17,23,28,34]. Follow-up times, where stated, were for the population of the study as a whole and may not have been the same for the sub-group of patients aged ≥75 years included in these meta-analyses. It is unlikely that studies with less than a year’s follow-up would have been long enough to capture effectiveness outcomes adequately. Finally, only two studies described follow-up of patients throughout the study and reported details about loss to follow-up [27,31].

Large RCTs have demonstrated that DOACs are at least as effective as warfarin for preventing stroke and systemic embolization in AF [36,37,38,39] and may also be associated with significantly less intracranial haemorrhage and major bleeding episodes [36,37,38,39]. A reduction in major or clinically relevant non-major bleeding with the DOAC apixaban was also observed for patients prescribed concomitant antiplatelet therapy following acute coronary syndrome or percutaneous coronary intervention [40]. Meta-analyses of RCTs found DOACs to be superior to warfarin for the prevention of stroke and systemic embolism [41,42,43,44]. However, many found no significant difference in the risk of major bleeding between DOACs and warfarin [41,42,43]. Concerns have been raised about the conduct of some of these trials and the impact this may have had on the validity of the results [45,46]. There is also less substantial evidence for the use of DOACs in older individuals, particularly those aged 75 years and over as people in this group only made up 30–40% of the total study populations [36,37,38,39]. Subgroup analyses of RE-LY [47], ROCKET-AF [48], and ENGAGE AF-TIMI 48 [49] and two meta-analyses [50,51] suggest that whilst at standard doses, the benefits of DOACs may be maintained in older patients, the risk of major extracranial, and in particular GI bleeding, is significantly higher. Meta-analyses which included both RCTs and observational studies identified similar trends to RCTs alone. On the whole, DOACs were favourable to warfarin in reducing stroke and systemic embolism [52,53], they found no significant difference between DOACs and warfarin for the outcome of major bleeding [53,54] and a significantly reduced risk of ICH [53]. Risk of GI bleeding was reported by only one study and found no significant difference in the pooled risk between DOACs and warfarin when RCTs or observational studies were combined [53].

In our meta-analysis, we used ischaemic stroke as the only effectiveness outcome as we argue that haemorrhagic stroke is a safety outcome. We found that the risk of ischaemic stroke was lower for DOACs. However, this was of borderline significance. It is likely that in the RCTs, the reduction in stroke with DOACs was due to their effect on haemorrhagic stroke rather than ischaemic stroke. Three meta-analyses of phase III RCTs showed this effect, all finding a significant reduction in the combined outcome of stroke and systemic embolism with DOACs, a significant reduction in haemorrhagic stroke but no significant difference in the risk of ischaemic stroke between DOACs and warfarin [42,43,44]. We found no significant difference in major bleeding between DOACs overall and VKAs which is in line with other meta-analyses in the age group [42,51]. Only Sharma and colleagues, who based their meta-analyses on individual DOACs, found that major bleeding was significantly reduced with apixaban compared with warfarin, which our results agreed with. We found rivaroxaban to significantly increase the risk of major bleeding whereas Sharma and colleagues found no significant difference. However, their meta-analyses included both trials of both AF and venous thromboembolism where we focused solely on AF [50]. We found the risk of GI bleeding was increased and intracranial haemorrhage decreased significantly with DOACs compared with warfarin, which is in keeping with previous meta-analyses of RCTs in patients aged over 75 years.

The results of these current meta-analyses and those published previously suggest that DOACs are no less effective than warfarin when prescribed to older patients with AF, do not significantly increase the risk of major bleeding as a whole, and cause significantly less intracranial haemorrhage than warfarin at the expense of an increased risk of gastrointestinal bleeding. However, the majority of data, both in the present meta-analyses and previous analyses, are based on sub-group or sensitivity analyses from studies that were not designed or powered to look at outcomes in this age group, so the results must be interpreted with caution. These meta-analyses highlight the high level of heterogeneity between the results of observational studies which may be due to a number of factors. Firstly, a number of studies were conducted in Asian patients [14,15,22,25,33] who are known to have a higher risk of ICH when treated with warfarin [55]; this may cause DOACs to appear more favourable for bleeding outcomes. Lower INR targets have also been advocated in Asian patients due to the increased risk of ICH and this may contribute to DOACs appearing more efficacious than warfarin [56]. Secondly, a number of studies included in these meta-analyses were conducted in the US [13,16,17,18,26,28,29,30,32,34]. The FDA did not approve the 110 mg dose of dabigatran which is widely used in Europe.

This systematic review has highlighted a need for research which is well conducted, adequately powered, and generalizable to older patients prescribed oral anticoagulants for AF. Observational studies should be transparent in how exposure is measured and use methods that address patients who switch or stop therapy during the study period. There are often differences in patient characteristics for those prescribed new medications to those prescribed older medications that influence prescribing. This potential for channelling is important and should be taken into account during the analysis. The follow-up time should be sufficient to assess all desired outcomes and follow-up should be described for all patients.

## Figures and Tables

**Figure 1 jcm-08-00554-f001:**
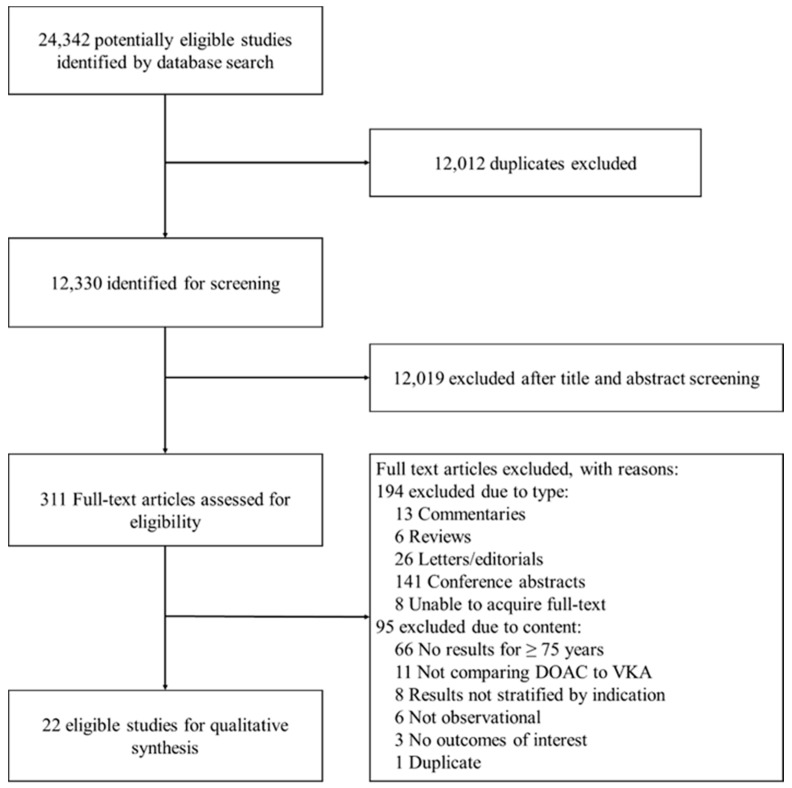
Study Selection.

**Figure 2 jcm-08-00554-f002:**
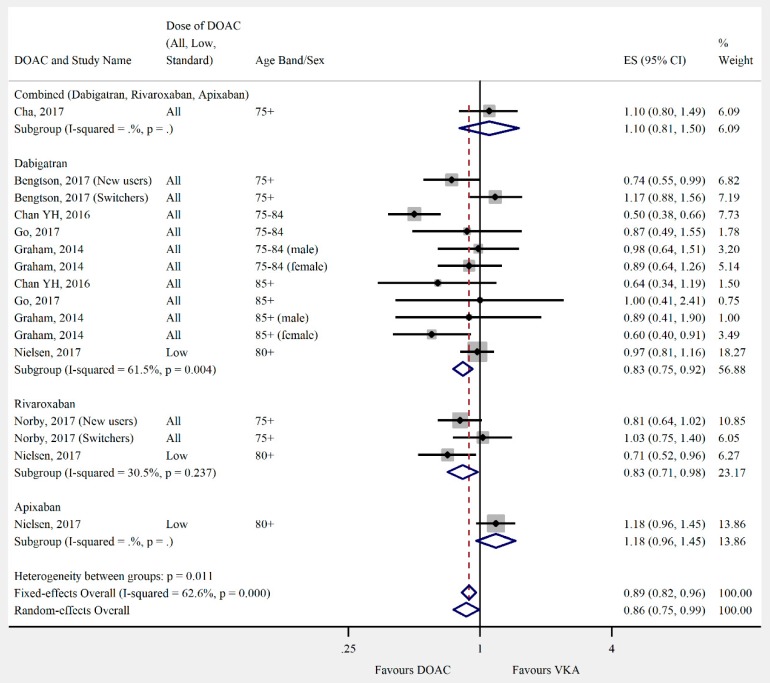
Meta-analysis of observational studies on ischaemic stroke stratified by direct oral anticoagulant (DOAC), then grouped by age band and DOAC dose. New users = no previous use of VKA, and switchers = previous use of VKA prior to starting DOAC. Effect sizes reported are hazard ratios. Sex is male and female unless otherwise stated.

**Figure 3 jcm-08-00554-f003:**
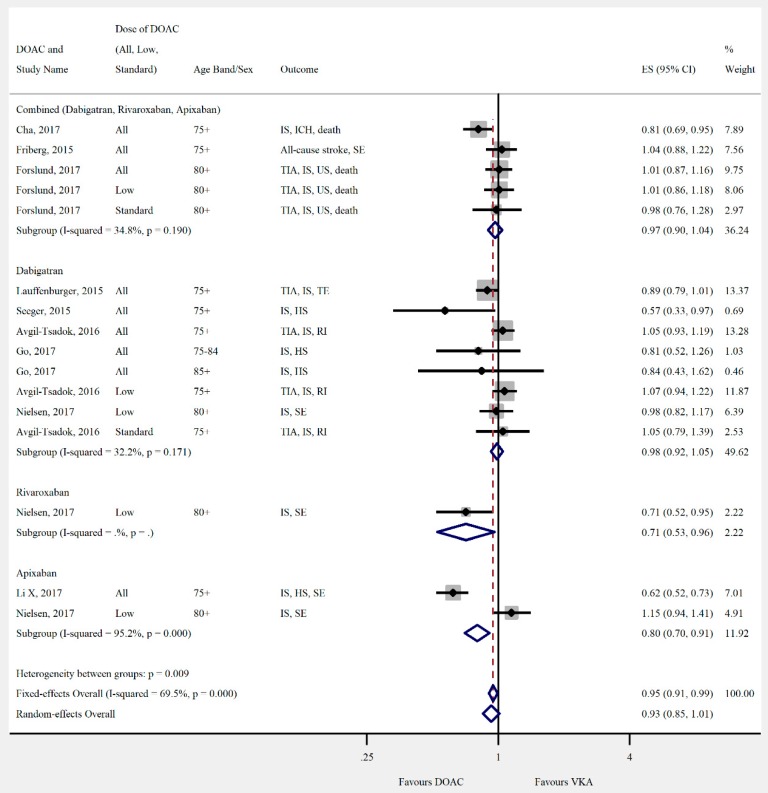
Meta-analysis of observational studies on composite effectiveness outcomes stratified by DOAC, then grouped by age band and DOAC dose. HS = haemorrhagic stroke, ICH = intracranial haemorrhage, IS = ischaemic stroke, RI = retinal infarct, SE = systemic embolism, TIA = transient ischaemic attack, and US = unspecified stroke. Effect sizes reported are hazard ratio. Sex is male and female unless otherwise stated.

**Figure 4 jcm-08-00554-f004:**
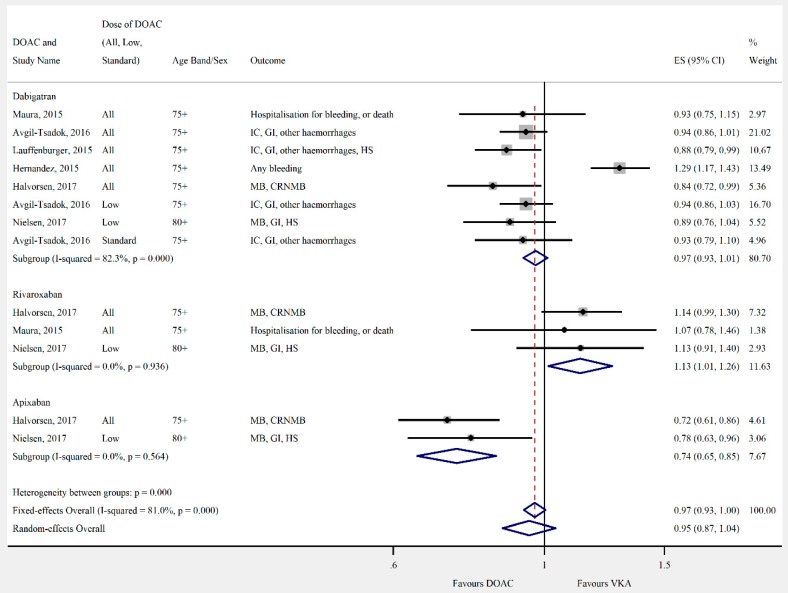
Meta-analysis of observational studies on composite safety outcomes stratified by DOAC, then grouped by age band and DOAC dose. CRNMB = clinically-relevant non-major bleeding, GI = gastrointestinal bleeding, HS = haemorrhagic stroke, IC = intracranial haemorrhage, and MB = major bleeding. Effect sizes reported are hazard ratio. Sex is male and female unless otherwise stated.

**Figure 5 jcm-08-00554-f005:**
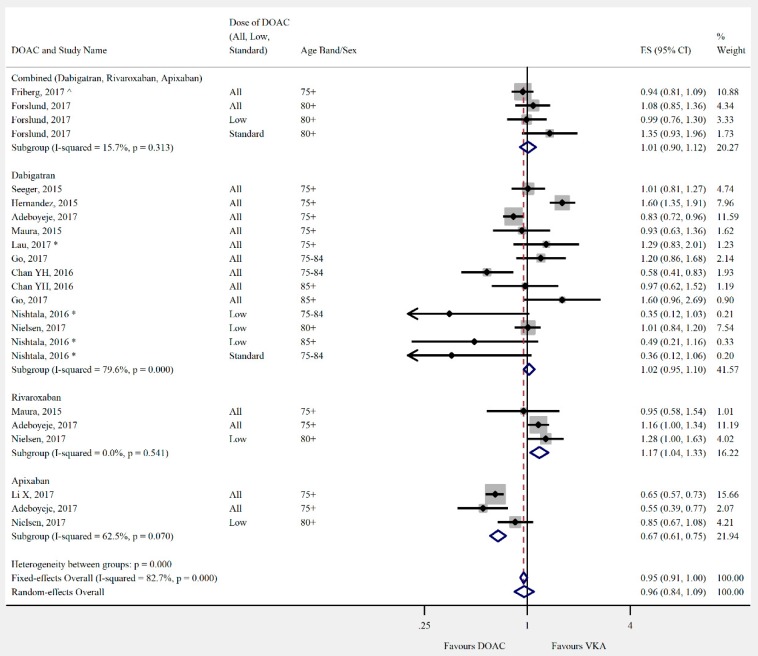
Meta-analysis of observational studies on major bleeding stratified by DOAC, then grouped by age band and DOAC dose. Effect sizes reported are hazard ratios, except where ^ = sub-hazard ratio, and * = incident rate ratio. Sex is male and female unless otherwise stated.

**Figure 6 jcm-08-00554-f006:**
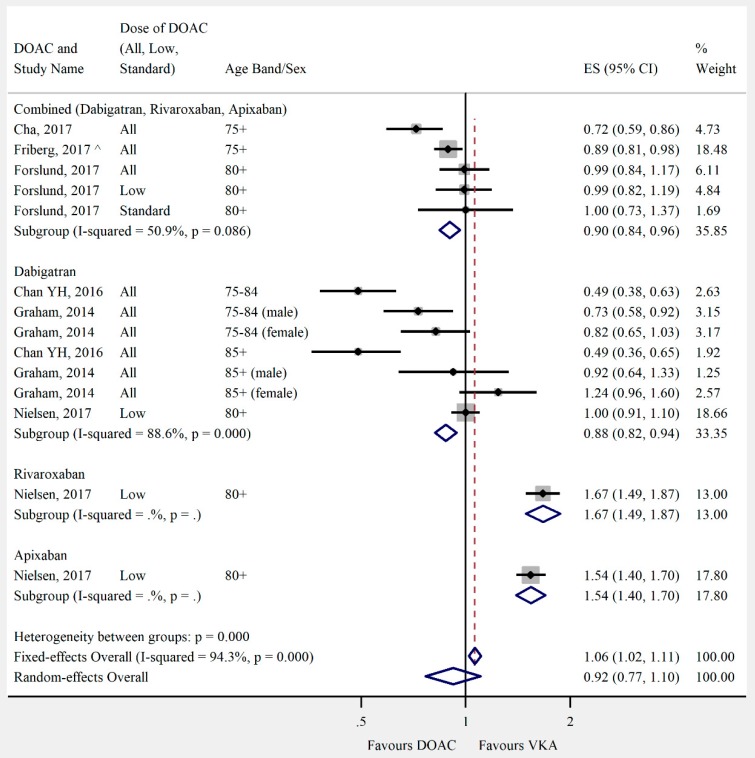
Meta-analysis of observational studies on mortality stratified by DOAC, then grouped by age band and DOAC dose. Effect sizes reported are hazard ratios, except where ^ = sub-hazard ratio. Sex is male and female unless otherwise stated.

**Figure 7 jcm-08-00554-f007:**
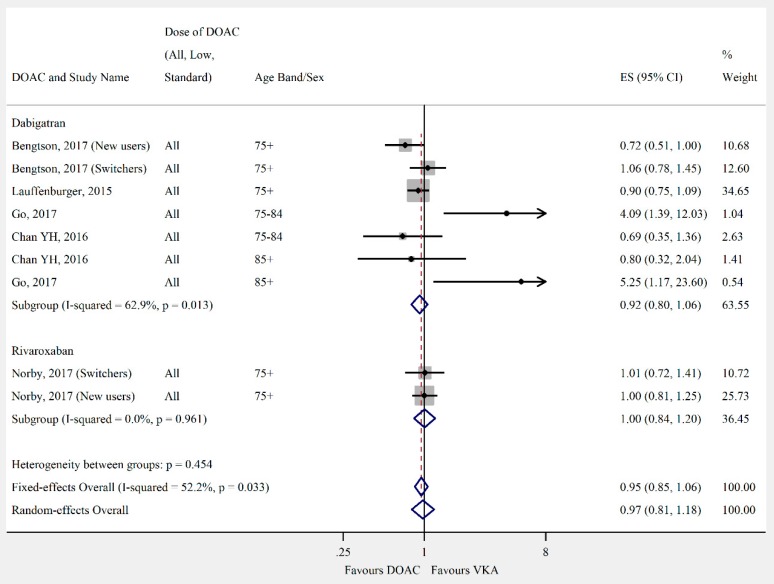
Meta-analysis of observational studies on myocardial infarction stratified by DOAC, then grouped by age band and DOAC dose. New users = no previous use of VKA, and switchers = previous use of VKA prior to starting DOAC. Effect sizes reported are hazard ratios. Sex is male and female unless otherwise stated.

**Figure 8 jcm-08-00554-f008:**
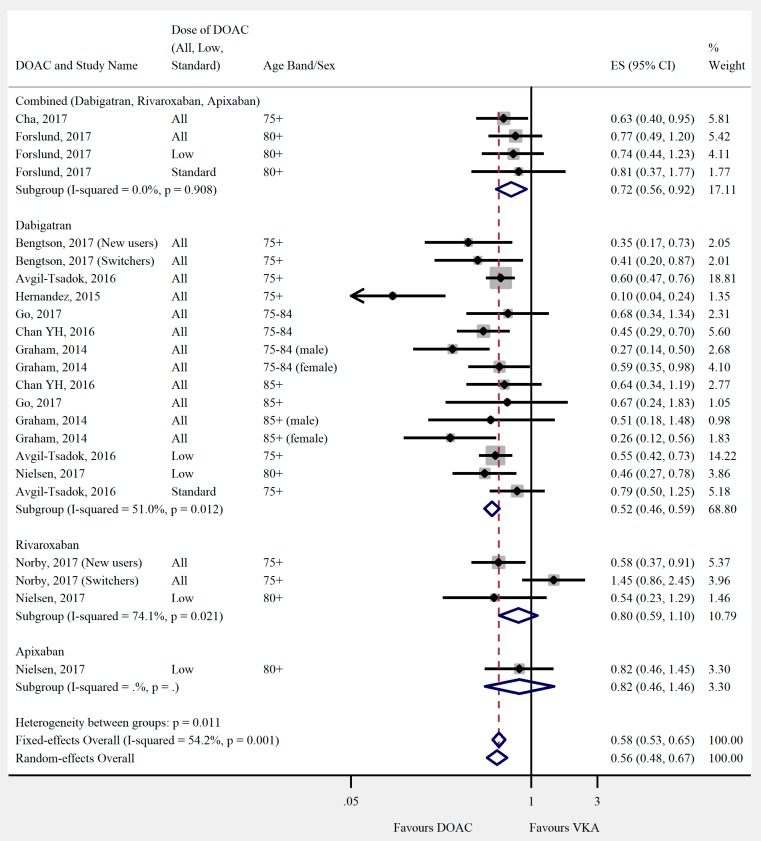
Meta-analysis of observational studies on intracranial haemorrhage stratified by DOAC, then grouped by age band and DOAC dose. New users = no previous use of VKA, and switchers = previous use of VKA prior to starting DOAC. Effect sizes reported are hazard ratios. Sex is male and female unless otherwise stated.

**Figure 9 jcm-08-00554-f009:**
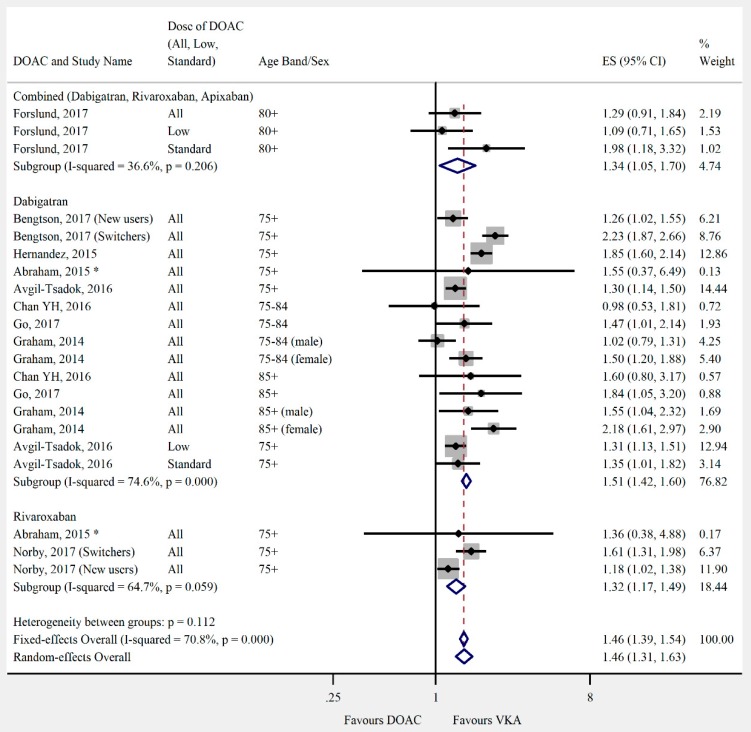
Meta-analysis of observational studies on gastrointestinal bleeding stratified by DOAC, then grouped by age band and DOAC dose. New users = no previous use of VKA, and switchers = previous use of VKA prior to starting DOAC. Effect sizes reported are hazard ratios, except where * = incident rate ratio. Sex is male and female unless otherwise stated.

**Table 1 jcm-08-00554-t001:** Quality assessment of included studies based on the modified Newcastle–Ottawa scoring system (Appendix C). Red = low quality, Amber = medium quality, and Green = high quality for each individual domain. Ordered by score (high to low) then alphabetically by author (A–Z).

	Score per Modified NOS Domain								Total	Comments
Author, Year	1	2	3	4	5	6	7	8		
Lau, 2017	2	1	2	1	2	1	2	0	11	
Halvorsen, 2017	2	1	2	0	2	1	2	0	10	No demonstration that outcome was not present at start of study.
Li X, 2017	2	1	2	0	2	1	2	0	10	No demonstration that outcome was not present at start of study.
Maura, 2015	2	1	1	0	2	1	2	1	10	Intention to treat approach stated so exposure assumed to continue from index date until censored. No demonstration that outcome was not present at start of study.
Friberg, 2017	2	1	1	0	2	1	1	1	9	Exposure monitoring after initial fill date not stated. No demonstration that outcome was not present at start of study.
Hernandez, 2015	1	1	2	0	2	1	2	0	9	Used a 5% sample of Medicare patients, so unclear how representative of US population as a whole. No demonstration that outcome was not present at start of study.
Chan YH, 2016	2	1	1	0	2	1	1	0	8	Exposure monitoring after initial fill date not stated. No demonstration that outcome was not present at start of study.
Forslund, 2017	2	1	1	0	2	1	1	0	8	No demonstration that outcome was not present at start of study.
Lauffenburger, 2015	1	1	1	0	2	1	2	0	8	Commercially insured population may not be truly representative of average US population. Exposure monitoring after initial fill date not stated. No demonstration that outcome was not present at start of study
Nielsen, 2017	2	1	1	0	2	1	1	0	8	Exposure monitoring after initial fill date not stated. No demonstration that outcome was not present at start of study.
Nishtala, 2016	1	1	2	0	2	1	1	0	8	Only includes patients with a hospital admission in the 5 years prior to study entry, so potentially represents sicker patients than in the average population. No demonstration that outcome was not present at start of study.
Norby, 2017	1	1	1	0	2	1	2	0	8	Exposure monitoring after initial fill date not stated. No demonstration that outcome was not present at start of study.
Seeger, 2015	1	1	2	0	2	1	1	0	8	Predominantly commercially insured patients only, so may not be truly representative of average US population. No demonstration that outcome was not present at start of study.
Abraham, 2015	1	1	2	0	2	1	0	0	7	Does not include Medicare patients, so may not represent older population well. Length of follow-up not stated, so unclear if long enough for outcomes to occur. No demonstration that outcome was not present at start of study.
Adeboyeje, 2017	1	1	2	0	2	1	0	0	7	Commercially insured patients only, so may not be truly representative of average US population. No demonstration that outcome was not present at start of study.
Avgil-Tsadok, 2016	1	1	1	0	2	1	1	0	7	Only represents patients diagnosed with AF as inpatients, so may not be truly representative. No demonstration that outcome was not present at start of study.
Bengtson, 2017	1	1	1	0	2	1	1	0	7	Exposure monitoring after initial fill date not stated. No demonstration that outcome was not present at start of study.
Cha, 2017	2	1	1	1	0	1	1	0	7	Intention to treat approach stated, so exposure assumed to continue from index date until censored. Propensity score used in analysis to adjust for confounders. However, it seems to include CHADS_2_-VASC_2_ score as the only variable.
Go, 2017	1	1	2	0	2	1	0	0	7	Predominantly privately insured population, so may not represent average US population.
Graham, 2014	1	1	2	0	2	1	0	0	7	Included only Medicare patients, so may not represent average US population. Follow-up duration not explicitly stated. However, sensitivity analysis for different lengths of follow-up did not affect results.
Kwon, 2016	0	1	1	0	0	2	1	0	5	Data from one hospital only. No description of how exposure was measured. Limited attempts to control for confounding. No demonstration that outcome was not present at start of study.
Chan PH, 2016	0	1	0	0	0	1	2	0	4	Data from one hospital only. No description of how exposure was measured. Limited attempts to control for confounding.
Modified Newcastle–Ottawa scale (NOS) domains:										
Selection 1. Representativeness of the exposed cohort 2. Selection of the non-exposed cohort 3. Ascertainment of exposure 4. Demonstration that outcome of interest was not present at start of study								Comparability 5. Comparability of cohorts on the basis of the design or analysis		Outcome 6. Assessment of outcome 7. Was follow-up long enough for outcomes to occur 8. Adequacy of follow-up of cohorts

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
