# Peer review of "Effectiveness and Safety of Direct Oral Anticoagulants versus Vitamin K Antagonists for People Aged 75 Years and over with Atrial Fibrillation: A Systematic Review and Meta-Analyses of Observational Studies"

_jcm, 2019, doi:10.3390/jcm8040554_

Round 1
Reviewer 1 Report
Dr Mitchell et al. investigated efficacy and safety of DOAC administration as compared to vitamin K antagonists in patients aged >= 75 years with AF. By conducting meta-analyses of the observation studies, they demonstrated a trend toward a lower risk of stroke in DOAC using, and no significant difference in major bleeding, mortality; or myocardial infarction in comparison with vitamin K antagonists.
Overall, the study was well-conducted, and was statistically well-analyzed with confidential results. I have no major comments on this manuscript. A few minor issues to be addressed.
-Since your study focused on elderly population, mean age and sex in the whole included subjects in this study would be represented. Moreover, how many subjects in total did you analyze in this study? Please describe.
-Introduction section is relatively long. It can be omitted (i.e., line 58-64).
Author Response
We thank Reviewer 1 for taking the time to review our manuscript and offer the following responses to the comments made:
Point 1: Since your study focused on elderly population, mean age and sex in the whole included subjects in this study would be represented. Moreover, how many subjects in total did you analyze in this study? Please describe
Response 1: The majority of data included in this review were from sub-group analyses and the demographics of the sub-group (such as mean age and sex) were often not described in the published papers. As we do not have access to patient level data from these studies we cannot calculate the mean age or sex in the whole included subjects in the study, although we agree this would have been a valuable addition to the paper if the data were available.
The total number of patients aged ≥ 75 years was not reported for 4 of the included studies, however, a sentence detailing the known number of subjects has been added from lines 161-163.
Point 2: Introduction section is relatively long. It can be omitted (i.e., line 58-64).
Response 2: Suggested lines have been removed.
Reviewer 2 Report
The authors have written a review assessing the safety and effectiveness of DOACs vs. VKAs for SPAF in the elderly population. The analysis appears well-done and comprehensive, though (and this is at no fault to the authors) the findings are not particularly novel.
Points to consider:
-Different guidelines have different "requirements" or thresholds for initiation of anticoagulation based on CHA2DS2VASc score. It is not correct to say that they advise initiation for a score of 2 or more. Some say a score of 1 or more not counting sexual category is appropriate for anticoagulation. Either be less prescriptivist about the cutoffs or discuss the different recommendations and cite each of them.
-I understand that discussion of DAPT plus anticoagulation for SPAF (so-called triple therapy) is somewhat beyond the scope of your article but I wonder if you should at least mention the AUGUSTUS trial, which was recently published, in your discussion. This was an RCT that included patients up to 95 years of age, and the results of this trial are in the process of completely changing the standard of care for SPAF in AFib in patients with ACS/PCI.
Author Response
We are very grateful to Reviewer 2 for their comments on our manuscript and offer the following responses:
Point 1: Different guidelines have different "requirements" or thresholds for initiation of anticoagulation based on CHA2DS2VASc score. It is not correct to say that they advise initiation for a score of 2 or more. Some say a score of 1 or more not counting sexual category is appropriate for anticoagulation. Either be less prescriptivist about the cutoffs or discuss the different recommendations and cite each of them.
Response 1: Thank you for highlighting this, we should have been clearer in what we meant. Lines 46-49 have been amended to be less prescriptive about the cut offs
Point 2: I understand that discussion of DAPT plus anticoagulation for SPAF (so-called triple therapy) is somewhat beyond the scope of your article but I wonder if you should at least mention the AUGUSTUS trial, which was recently published, in your discussion. This was an RCT that included patients up to 95 years of age, and the results of this trial are in the process of completely changing the standard of care for SPAF in AFib in patients with ACS/PCI.
Response 2: As acknowledged by the reviewer, this trial is beyond the scope of this article. Whilst the trial protocol states that patients up to 95 years of age were eligible for inclusion, no data is presented for outcomes specifically in the older patients. However, we agree that this trial is an important addition to the body of evidence for the use of DOACs in AF so a reference to the reduction in major bleeding episodes with apixaban identified in the AUGUSTUS trial has been added to the discussion from line 309-311 and a citation added.